# Spermidine from arginine metabolism activates Nrf2 and inhibits kidney fibrosis

Seishi Aihara[1], Kumiko Torisu [1,2 ✉], Yushi Uchida[1], Noriyuki Imazu[1], Toshiaki Nakano [1,3 ✉] & Takanari Kitazono[1]

Kidney metabolism may be greatly altered in chronic kidney disease. Here we report that arginine metabolism is the most altered in unilateral ureteral obstruction (UUO)-induced fibrosis of the kidneys in metabolomic analysis. Spermidine is the most increased metabolite of arginine. In human glomerulonephritis, the amount of spermidine shown by immunostaining is associated with the amount of fibrosis. In human proximal tubule cells, spermidine induces nuclear factor erythroid 2-related factor 2 (Nrf2). Subsequently, fibrotic signals, such as transforming growth factor β1 secretion, collagen 1 mRNA, and oxidative stress, represented by a decrease in the mitochondrial membrane potential is suppressed by spermidine. UUO kidneys of *Arg2* knockout mice show less spermidine and significantly exacerbated fibrosis compared with wild-type mice. Nrf2 activation is reduced in *Arg2* knockout UUO kidneys. Spermidine treatment prevents significant fibrotic progression in *Arg2* knockout mice. Spermidine is increased in kidney fibrosis, but further increases in spermidine may reduce fibrosis.

[1] Department of Medicine and Clinical Science, Graduate School of Medical Sciences, Kyushu University, Fukuoka, Japan. [2] Department of Integrated Therapy for Chronic Kidney Disease, Graduate School of Medical Sciences, Kyushu University, Fukuoka, Japan. [3] Center for Cohort Studies, Graduate School of Medical Sciences, Kyushu University, 3-1-1 Maidashi, Higashi-ku, Fukuoka, Japan. ✉email: torisu.kumiko.350@m.kyushu-u.ac.jp; nakano.toshiaki.455@m.kyushu-u.ac.jp

Kidney fibrosis is the final common pathway for most types of progressive kidney diseases and eventually leads to end-stage kidney disease[1]. Chronic kidney disease (CKD) with kidney fibrosis is associated with complications, such as cardiovascular disease, anemia, inflammation, and malnutrition[2]. Therefore, not only systemic metabolism, but also kidney metabolism, may be greatly altered in patients with CKD. Recent advances in omics analysis have reported that fatty acid[3] or lysine[4] metabolism is altered in renal injury. The human kidney plays an important role in maintaining homeostasis of amino acid pools throughout the body through synthesis, degradation, filtration, reabsorption, and urinary excretion of amino acids and peptides. The kidney is the major excretion site for glutamine and proline, as well as the production site for serine, cysteine, and arginine[5]. The balance between amino acid excretion and metabolism is considered important for renal protection[6].

Arginine is not only a material for protein synthesis, but also a precursor for urea, nitric oxide (NO)[7], polyamines, proline, glutamic acid, creatine, and agmatine[8]. Therefore, arginine is involved in a variety of biological processes. L-arginine is the substrate for nitric oxide synthase (NOS) and also the substrate for arginase[7]. Kidney mass reduction by uninephrectomy causes changes in arginine metabolism and increases blood pressure[9]. Arginase hydrolyzes L-arginine into urea and L-ornithine, and ornithine-derived polyamines are essential for the growth and function of cells[10]. Spermidine (Spd) is an ornithine-derived polyamine. One of five conserved metabolites released from apoptotic cells is Spd. Apoptotic metabolites are selectively released as "good-bye" signals to suppress inflammation in surrounding cells, leading to wound healing[11]. Arginine metabolism in the kidney may be important because arginase 2 (ARG2) is predominantly expressed in renal tubular epithelial cells (RTECs)[12,13]. Our previous study showed that ischemia–reperfusion injury in the kidney was attenuated in Arg2 knockout (KO) mice or arginase inhibitor-treated mice through regulating nitrosative stress[12]. We have previously investigated the importance of arginine metabolism in acute kidney injury, but not in chronic kidney injury. Therefore, in this study, we investigated the role of arginine metabolism involving ARG2 and its metabolites in kidney fibrosis using comprehensive metabolic analysis. We used mice with unilateral ureteral obstruction (UUO) as a model of CKD. The fibrotic inhibitory effect of Spd was investigated.

## Results

**Arginine metabolism is upregulated in the UUO kidney in mice.** To examine metabolic changes in UUO, we compared metabolites from UUO kidneys and sham-operated kidneys. "Arginine and proline metabolism" and "arginine biosynthesis" were most altered when pathway analysis was performed on significantly increased metabolites in the UUO kidney among 110 detected amino acids (Fig. 1a and Table 1). When we focused on "arginine and proline metabolism", we found that ornithine and proline concentrations were significantly higher (>2 times, both $P < 0.01$) in the UUO kidney than in the sham-operated kidney (Fig. 1b, c). With regard to ornithine-derived polyamines, putrescine and Spd concentrations were approximately twice as high in the UUO kidney as those in the sham-operated kidney, while spermine (Spm) concentrations showed no difference (Fig. 1b). Although arginine metabolism was activated, the substrate L-arginine was not decreased in UUO kidneys, but increased instead (Fig. 1b). A schematic diagram of "arginine biosynthesis" and "arginine and proline metabolism" is shown in Fig. 1c.

**Spermidine increases in response to kidney fibrosis.** We focused on Spd, which was the most increased arginine metabolite. Spd was fifth among the 110 measured metabolites that was specifically increased in the UUO kidney relative to controls (Table 2). Immunostaining showed that Spd levels were remarkably increased in the UUO kidney, especially in the tubules (Fig. 2a, b). Western blot analysis of mouse UUO kidneys showed that ARG2 protein levels were approximately two times higher in the UUO kidney relative to the sham-operated kidney (Fig. 2c, d). Immunostaining of ARG2 was significantly greater in the UUO kidney than in the sham-operated kidney and was localized mostly in RTECs (Fig. 2e, f, $P < 0.05$). We hypothesized that Spd levels are correlated with the severity of interstitial fibrosis in the human kidney. Human kidney specimens from donor kidneys at the time of living donor kidney transplantation as a control or kidneys of IgA nephropathy were evaluated. The characteristics of patients at diagnosis by renal biopsy are shown in Table 3. The tubular atrophy/interstitial fibrosis score (T score) was significantly associated with higher urinary protein and kidney function. Spd levels were remarkably enhanced in the kidney with fibrosis, especially in the tubules (Fig. 2g, h). A toxic compound produced from Spm, acrolein, was also increased in the UUO kidney (Supplementary Fig. S1a, b). The expression of spermine oxidase (Smox) mRNA, which generates Spd from Spm, was also significantly higher in the UUO kidney than in the sham-operated kidney (Supplementary Fig. S1c, $P < 0.01$). This finding is consistent with the finding that only Spm among the polyamines was not increased in the UUO kidney in metabolomic analysis (Fig. 1b). These findings indicate that arginine metabolism is altered by fibrotic stimuli in mice.

**Oxidative stress upregulates ARG2 protein levels and polyamine including spermidine in RTECs.** We focused on ARG2 as an important enzyme that regulates arginine biosynthesis. During the fibrotic process, tubules are expected to be exposed to a large amount of reactive oxygen species stress. Therefore, HK-2 cells were exposed to 500 μM hydrogen peroxide ($H_2O_2$) for 24 h. Oxidative stress increased ARG2 protein levels in HK-2 cells (Fig. 3a, b). This result is consistent with the enhanced ARG2 protein levels in the UUO kidney, especially in RTECs. We then evaluated changes in the production of polyamine or Spd during oxidative stress. Polyamine red staining showed that polyamine production was induced by $H_2O_2$ (Fig. 3c). ARG2 levels were reduced to ~10% of control levels in HK-2 cells transfected with Arg2-siRNA (Supplementary Fig. S2a). Polyamine was reduced by Arg2 knockdown in HK-2 cells (Fig. 3c, d). This suppression of polyamine production in Arg2 knockdown cells was observed even in the absence of oxidative stress. In an immunofluorescence study, Spd formed puncta, which were diffusely distributed in the cytoplasm (Fig. 3e). Overexpression of ARG2 (Supplementary Fig. S2b) increased the amount of Spd, while knockdown of Arg2 decreased the amount of Spd (Fig. 3e, f). These findings suggest that the amount of Spd is largely dependent on ARG2 expression levels.

**Spd activates the transcription factor nuclear factor erythroid 2-related factor 2 in RTECs.** We hypothesized that Spd plays an important role in the development of kidney fibrosis. Initially, cell survival during 24-h exposure of Spd to tubular cells was investigated (Supplementary Fig. S3a). Below a concentration of 20 μM, cell numbers did not change, but above 100 μM, cell death occurred. Therefore, HK-2 cells were exposed to 20 μM Spd for 24 h to increase Spd levels in cells. HK-2 cells incubated with Spd showed high Spd levels in the cytoplasm (Supplementary Fig. S3b). Surprisingly, Spd caused activation of the transcription factor nuclear factor erythroid 2-related factor 2 (Nrf2), which was increased by ~15 times that of controls (Fig. 4a, b). There was no significant change in protein levels of kelch-like ECH-associated protein 1 (Keap1), which is an

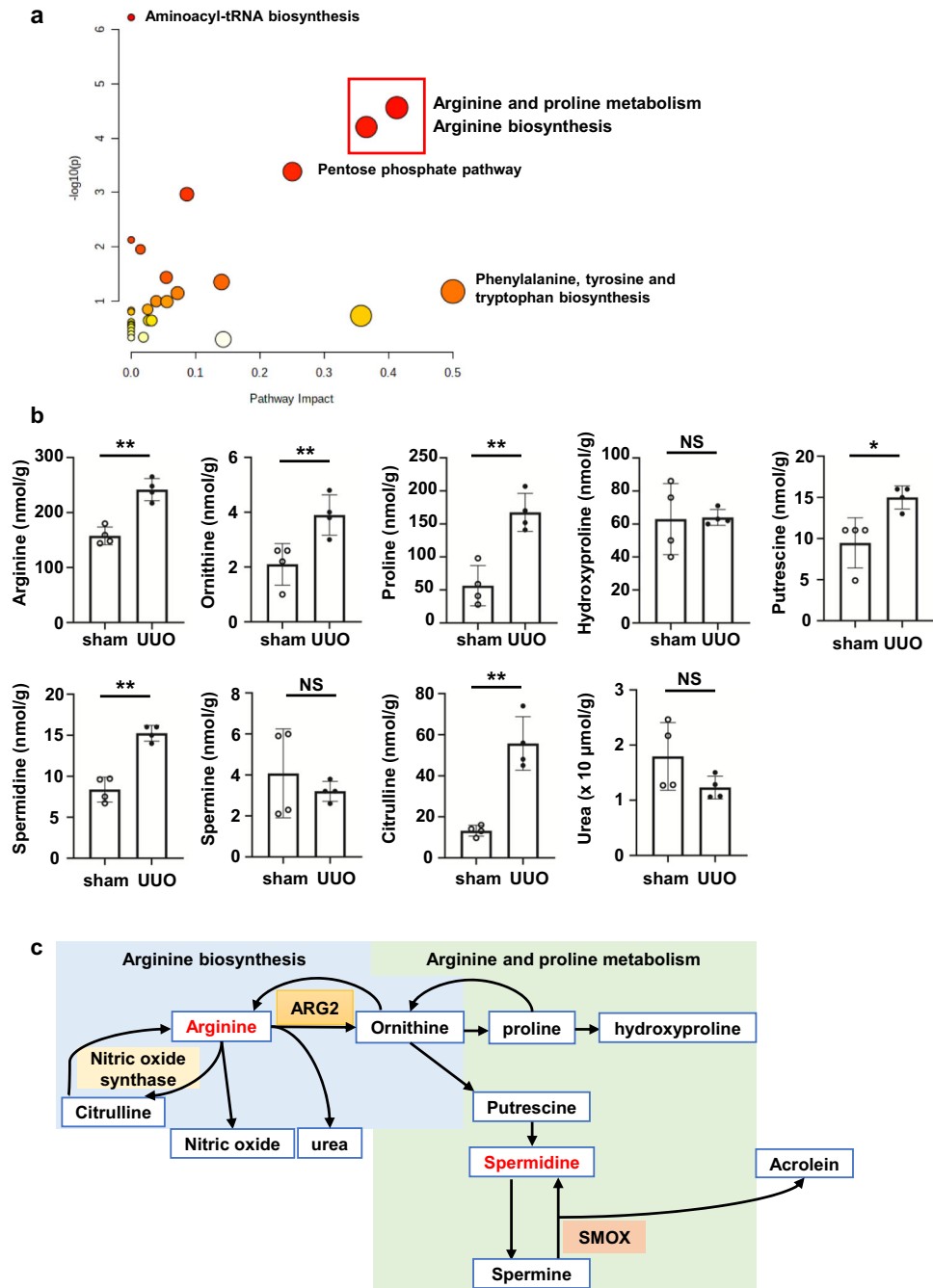

**Fig. 1 Arginine metabolism is upregulated in the UUO kidney of mice. a** Results of pathway analysis using MetaboAnalyst. Of the 110 metabolites, pathway analysis was performed only for metabolites with a fold change of ≥1.5 in the UUO kidney relative to shams ($n = 4$ in each group). Red squares indicate "arginine and proline metabolism" and "arginine biosynthesis". **b** Absolute quantitative values of metabolites related to arginine metabolites in the UUO kidney compared with those in shams ($n = 4$ in each group). Data are indicated as means ± SD. **c** Schematic diagram of "arginine biosynthesis" and "arginine and proline metabolism". *$P < 0.05$, **$P < 0.01$. NS not significant, UUO unilateral ureteral obstruction, ARG2 arginase 2, SMOX spermine oxidase.

adapter of the ubiquitin ligase complex that targets Nrf2 in HK-2 cells (Fig. 4a, c). Although the activation of Nrf2 was expected to be associated with the activation of inflammatory signaling, there was no significant change in the phosphorylation of nuclear factor-κB (NF-κB) by Spd in HK-2 cells (Fig. 4d, e). Immuno-cytochemistry showed the nuclear translocation of Nrf2 by Spd in HK-2 cells (Fig. 4f, g). Consistent with the nuclear translocation of Nrf2, the expression of Nrf2-targeted genes, *HO-1*, *NQO1*, and *GCLM* mRNA, was induced in HK-2 cells by Spd (Fig. 4h–j). These data suggest that Spd activates the transcription factor Nrf2 in RTECs.

**Spd activates autophagy, promoting dissociation of Keap1 from Nrf2.** We then investigated the detailed mechanism by which Spd activates Nrf2 in HK-2 cells. Polyamines, including Spd, are potent inducers of autophagy[14]. While Spd increased conversion of microtubule-associated protein 1 A/1B-light chain 3 (LC3)-I to LC3-II, the amount of p62, which is a typical substrate for autophagy, did not decrease, but increased instead (Fig. 5a–c). Although there was no significant difference in Keap1 protein levels between the control and Spd-treated HK-2 cells (Fig. 4c), co-localization of Keap1 and p62 was significantly increased by Spd (Fig. 5d, e, $P < 0.05$). Consistent with the

**Table 1 Network enrichment analysis results from metabolites in unilateral ureteral obstruction.**

| | Total | Expected | Hits | Raw P value | Impact |
|---|---|---|---|---|---|
| Arginine and proline metabolism | 38 | 0.65604 | 6 | 2.74E-05 | 0.41283 |
| Arginine biosynthesis | 14 | 0.2417 | 4 | 6.23E-05 | 0.36548 |
| Pentose phosphate pathway | 22 | 0.37981 | 4 | 0.000414 | 0.25044 |
| Alanine, aspartate and glutamate metabolism | 28 | 0.4384 | 4 | 0.00108 | 0.08654 |
| Glutathione metabolism | 28 | 0.4384 | 3 | 0.011244 | 0.01438 |
| Fructose and mannose metabolism | 18 | 0.31076 | 2 | 0.037018 | 0.05452 |
| Citrate cycle (TCA cycle) | 20 | 0.34529 | 2 | 0.045012 | 0.14041 |
| Phenylalanine, tyrosine and tryptophan biosynthesis | 4 | 0.069057 | 1 | 0.067355 | 0.5 |
| Glycolysis/gluconeogenesis | 26 | 0.44887 | 2 | 0.072294 | 0.07202 |
| Purine metabolism | 66 | 1.1394 | 3 | 0.10193 | 0.0389 |
| Glyoxylate and dicarboxylate metabolism | 32 | 0.55246 | 2 | 0.10364 | 0.05556 |
| Pyrimidine metabolism | 39 | 0.67331 | 2 | 0.14399 | 0.0256 |
| Phenylalanine metabolism | 12 | 0.20717 | 1 | 0.18922 | 0.35714 |
| Butanoate metabolism | 15 | 0.25896 | 1 | 0.23084 | 0.03175 |
| Starch and sucrose metabolism | 15 | 0.25896 | 1 | 0.23084 | 0.02703 |
| Glycerophospholipid metabolism | 36 | 0.62151 | 1 | 0.46975 | 0.01896 |
| Tryptophan metabolism | 41 | 0.70784 | 1 | 0.51507 | 0.14305 |

*UUO* unilateral ureteral obstruction, *TCA cycle* tricarboxylic acid cycle.
Network analysis was performed by focusing on metabolites that were increased above 1.0 in the UUO/sham ratio and had a P value of ≤0.05 with the Kyoto Encyclopedia of Genes and Genomes database.

**Table 2 Top ten metabolites that were significantly increased in unilateral ureteral obstruction model compared with those in shams.**

| Order | Compound name | Ratio (UUO/sham) | P value |
|---|---|---|---|
| 1 | L-Lysine | 1.963063 | 2.21E-07 |
| 2 | 6-Phosphogluconic acid | 3.066324 | 3.14E-05 |
| 3 | γ-aminobutyric acid | 2.928807 | 0.000221 |
| 4 | L-Tryptophan | 1.658721 | 0.000254 |
| 5 | Spermidine | 1.823254 | 0.000469 |
| 6 | Phosphoribosyl pyrophosphate | 2.298372 | 0.000477 |
| 7 | L-Arginine | 1.530962 | 0.000701 |
| 8 | L-Isoleucine | 1.700839 | 0.000777 |
| 9 | Fructose 1,6-diphosphate | 66.64083 | 0.001271 |
| 10 | L-Phenylalanine | 1.790486 | 0.001512 |

*UUO* unilateral ureteral obstruction.

increased co-localization of Keap1 and p62, the co-localization of Keap1 with LC3-positive autophagosomes increased in Spd-treated cells (Supplementary Fig. S4a). These results suggest that Keap1 translocates to autophagosomes by binding to p62. To assess autophagic flux, LC3-II levels in cells were examined in the presence of 100 μM hydroxychloroquine (HCQ). LC3-II levels were remarkably increased (Fig. 5f, g) in the presence of HCQ. These results suggest that Spd enhances autophagic flux and simultaneously activates degradation in lysosomes. Furthermore, p62, which was increased by Spd, was not further increased by the addition of HCQ (Fig. 5f, h). The activation of Nrf2 (Fig. 5i) and the expression of target genes, such as *HO-1* and *GCLM* (Fig. 5j, k), were partially reduced by HCQ. Significant Nrf2 activation by Spd was suppressed to two-thirds of normal levels in *Atg5* knockdown cells (Supplementary Fig. S4b, c; $P < 0.01$, which changed to not significant after knockdown), which was accompanied by a trend towards reduced *HO-1* mRNA expression (Supplementary Fig. S4d). A similar trend was observed in *Atg5* KO mouse embryonic fibroblasts, although the activation of Nrf2 by Spd was not as strong as that in tubular cells (Supplementary Fig. S4e–g). Nrf2 activation by Spd is expected to be mediated in part through autophagy.

**Spd has antifibrotic effects, but does not inhibit the endothelin pathway like bardoxolone methyl.** To further investigate whether Spd has a protective effect on fibrotic signaling in tubular cells, transforming growth factor β1 (TGFβ1), which is a fibrotic signal, secreted from HK-2 cells was measured with or without Spd. A certain amount of TGFβ1 was released from HK-2 cells, but the addition of Spd significantly suppressed this release (Fig. 6a, $P < 0.01$). Collagen 1 mRNA expression in tubular cell fibrosis induced by TGFβ1 was significantly suppressed by Spd (Fig. 6b, $P < 0.01$). TGFβ1 stimuli alone did not increase *HO-1* mRNA expression. Spd induced *HO-1* gene expression even in the presence of TGFβ1 (Fig. 6c). We investigated whether Spd inhibits oxidative stress because oxidative stress accelerates the progression of fibrosis in the kidney. Oxidative stress, which was represented by a decrease in mitochondrial membrane potential caused by $H_2O_2$, was significantly suppressed by the addition of Spd (Fig. 6d, e). These results suggest that Spd plays a protective role in fibrotic signaling in RTECs. The effect of Spd on the endothelin pathway was then examined. Bardoxolone methyl (CDDO-me), which is a representative Nrf2 inducer, increases the incidence of heart failure via inhibition of the endothelin pathway[15]. Real-time polymerase chain reaction (PCR) showed that CDDO-me 40 nM and Spd 20 μM were equivalent in their induction of *HO-1* mRNA expression (Fig. 6f). CDDO-me concentrations in the medium were determined with reference to the patient's blood concentrations[16]. Endothelin-1 (*ET-1*) mRNA expression and ET-1 protein concentrations were suppressed with CDDO-me, but not with Spd (Fig. 6g, h).

**UUO-induced kidney fibrosis is aggravated and Nrf2 is decreased in the *Arg2* KO kidney.** To determine whether ARG2 is involved in kidney fibrosis, we used the UUO model with *Arg2* KO mice. There was no significant difference in body weight or kidney function between wild-type (WT) and *Arg2* KO mice with UUO. Systolic blood pressure in *Arg2* KO mice with UUO tended to be higher than that in WT mice, but this was not significant (Supplementary Table S1). Spd protein levels were remarkably reduced in the *Arg2* KO UUO kidney (Fig. 7a, b). Interstitial fibrosis of the kidney determined by Sirius red or Masson trichrome staining was significantly aggravated in *Arg2* KO mice compared with that in WT mice (Fig. 7c–f, $P < 0.05$). Collagen 1 protein levels were significantly higher in *Arg2* KO mice than in

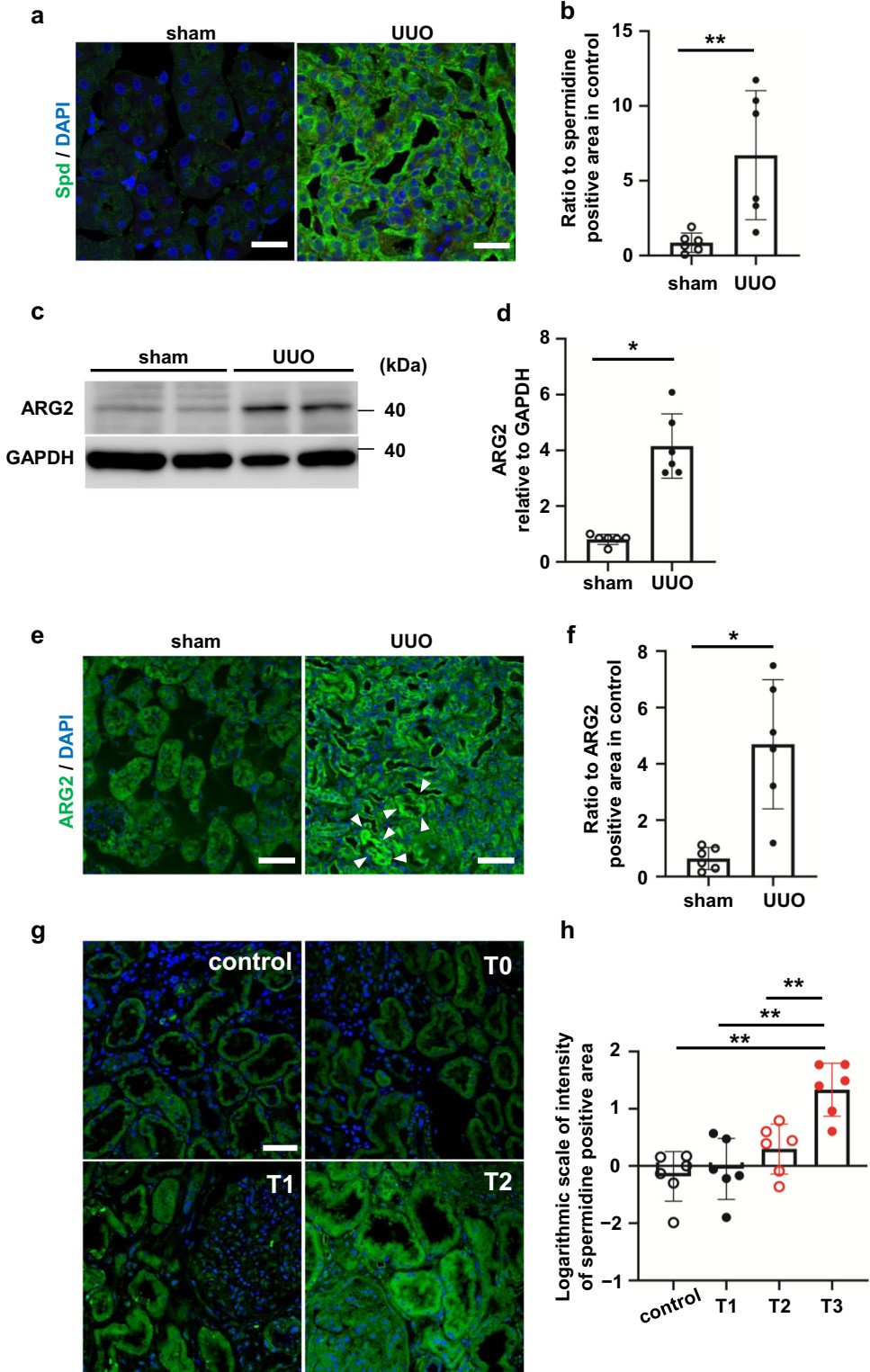

WT mice (Fig. 7g, $P < 0.01$). In *Arg2* KO mice, α smooth muscle actin (αSMA) protein levels in the *Arg2* KO kidney tended to be increased (Fig. 7h). Mature TGFβ protein levels were also increased in *Arg2* KO mice (Fig. 7i). There was no difference in L-arginine concentrations between WT and *Arg2* KO mice (Supplementary Fig. S5). Protein levels of ARG1, which is another arginase isoform, were not increased in the *Arg2* KO UUO kidney (Supplementary Fig. S6a, b). In addition to a decrease in Spd, *Smox* mRNA expression and SMOX protein levels

were significantly decreased in *Arg2* KO mice (Supplementary Fig. S7a–c, both $P < 0.05$). Acrolein protein levels were not obviously increased in *Arg2* KO mice. (Supplementary Fig. S7d, e). In *Arg2* KO mice, the expression of enzymes related to arginine metabolism other than ARG2 was also altered. Protein levels of endothelial NOS (eNOS) and the endothelial marker CD31 were significantly increased in the *Arg2* KO UUO kidney (Supplementary Fig. S8a–c, both $P < 0.01$), which suggested that angiogenesis was associated with fibrosis. On the basis of the

**Fig. 2 UUO increases spermidine and ARG2 protein levels in the mouse kidney. a** Confocal immunofluorescence microscopic images of Spd in control and UUO kidneys. Green, anti-Spd antibody; blue, DAPI. Scale bars, 50 µm. **b** Quantification of the Spd-positive area in control and UUO kidneys ($n = 6$ in each group). Data are indicated as means ± SD. **c** Western blot analysis of ARG2 protein levels in the whole kidney of sham and UUO mice. **d** Relative levels of ARG2 protein normalized to GAPDH in sham and UUO kidneys are shown ($n = 6$ in each group). Data are indicated as means ± SD. **e** Immunohistochemistry of ARG2 in control and UUO kidneys. Green, anti-ARG2 antibody; blue, DAPI. Scale bars, 50 µm. White arrowheads indicate ARG2-positive tubules. **f** Quantification of the ARG2-positive area in sham and UUO kidneys ($n = 6$ in each group). Data are indicated as means ± SD. $*P < 0.05$, $**P < 0.01$. **g** Immunostaining of Spd in the human kidney. Control, donor kidneys; T0–T2, interstitial fibrosis/tubular atrophy scores from the Oxford classification of IgA nephropathy. The percentages of lesions in the cortical area were as follows: T0 = 0%–25%, T1 = 26%–50%, and T2 = >50%. Scale bars, 50 µm. **h** Quantification of the Spd-positive area in human kidneys ($n = 6$ in each group). Data are indicated as means ± SD. $**P < 0.01$. Spd spermidine, UUO unilateral ureteral obstruction, ARG2 arginase 2, DAPI 4′,6-diamidino-2-phenylindole, GAPDH glyceraldehyde-3-phosphate dehydrogenase.

| Table 3 Patients' characteristics at the time of a renal biopsy in relation to the degree of fibrosis. | | | | | |
|---|---|---|---|---|---|
| Grade | Control ($n = 6$) | T0 ($n = 6$) | T1 ($n = 6$) | T2 ($n = 6$) | *P* for trend |
| Age, years | 52.5 ± 12.4 | 52.2 ± 13.3 | 51.8 ± 15.0 | 52.2 ± 15.4 | 0.94 |
| Sex, male, % | 50 | 50 | 50 | 50 | 1 |
| U-P/U-Cr ratio, g/gCr | 0.04 ± 0.01 | 1.26 ± 0.87 | 2.11 ± 1.26 | 2.26 ± 1.03 | <0.01 |
| eGFR, mL/min/1.73 m$^2$ | 83.7 ± 4.42 | 59.4 ± 17.2 | 52.6 ± 18.3 | 45.1 ± 19.4 | <0.01 |

*Cr* creatinine, *eGFR* estimated glomerular filtration rate, *U* urinary.
Data are expressed as the mean ± standard deviation or number (%).

above-mentioned results, inflammation and oxidative stress are expected to be enhanced in the *Arg2* KO UUO kidney. However, Nrf2 protein levels in the *Arg2* KO UUO kidney were not increased, but decreased by two-thirds those of the WT UUO kidney (Fig. 7j, k). Accordingly, *HO-1* mRNA expression was significantly reduced in the *Arg2* KO UUO kidney (Fig. 7l, $P < 0.01$). Consistent with the results of the cell experiments, the lack of Spd in *Arg2* KO mice may inhibit the activation of Nrf2. On the basis that *Arg2* KO mice had reduced Spd levels, we hypothesized that fibrosis is exacerbated by inadequate Nrf2 activation in *Arg2* KO mice. We investigated whether Spd supplementation suppresses fibrosis. Kidney fibrosis, which was significantly higher in *Arg2* KO mice than in WT mice, was suppressed by treatment with Spd and the significant difference between WT and *Arg2* KO mice disappeared (Fig. 7m, n). In the *Arg2* KO kidney, *HO-1* mRNA expression of the Nrf2 target gene was significantly reduced, but tended to increase with Spd treatment (Supplementary Fig. S9).

## Discussion

In recent years, kidney injury and energy metabolism have been popular areas of focus in drug discovery. Amino acid metabolism is activated in the mouse diabetic kidney model[17] or in hypertensive kidney damage[4], but few studies have investigated whether metabolites are involved in kidney disorders. To the best of our knowledge, this study shows that the metabolite Spd is involved in antifibrosis.

Although arginine metabolism was expected to be impaired in *Arg2* KO mice, there was no difference in L-arginine concentrations between WT and *Arg2* KO UUO kidneys (Supplementary Fig. S5a). Parallel to the exacerbation of fibrosis, eNOS and CD31 protein levels were increased in the *Arg2* KO UUO kidney (Supplementary Fig. S8a–c). L-arginine, which accumulates without undergoing degradation in *Arg2* KO mice, may be degraded by angiogenesis-induced eNOS. In UUO kidneys in our study, ornithine (metabolite of ARG2) was increased 1.9 fold, whereas citrulline (metabolite of NOS) was increased 4.2-fold (Fig. 1b). NO production modifies renal hemodynamics in the early phase of UUO[18] and plays a protective role against fibrosis in the chronic phase of UUO[19,20]. In a previous

study, acute ischemia–reperfusion injury in *Arg2* KO mice was reduced compared with that in WT mice in contrast to the present study[12]. There was no increase in eNOS, inducible NOS, or neuronal NOS expression in the *Arg2* KO kidney with acute injury compared with that in the WT[12]. The differences in NOS expression changes may have led to phenotypic differences in acute and chronic renal injury in our study because eNOS is increased in the UUO kidney.

In recent years, there have been many important reports on the benefits of polyamines in pathological mouse models. Increased colonic luminal polyamines promote longevity in mice[21], and bacterial-derived polyamines ameliorate symptoms in colitis model mice[22]. Oral supplementation of Spd extends the lifespan of mice and exerts cardioprotective effects in old mice[23]. In our UUO model, Spd administration also reduced fibrosis. SMOX is an enzyme that generates Spd and the highly toxic compound acrolein from Spm. A previous study showed that Spm concentrations were decreased with high SMOX activity, and plasma acrolein concentrations were increased in patients with CKD[24]. Acrolein was found to increase with renal ischemia-reperfusion injury and induce tubular cell death[25]. Cisplatin-induced acute kidney injury is alleviated in *Smox* KO mice in which Spd is reduced[26]. Chronic kidney injury may also be promoted because it is exacerbated in *Smox* KO mice in the chronic inflammatory model of dextran sulfate sodium-induced enteritis[27]. Not only a lack of ARG2, but also a decrease in SMOX, may contribute to the low Spd expression in *Arg2* KO mice.

In a lipopolysaccharide-induced acute kidney injury model, Spd inhibited inflammasome activation and promoted mitochondrial respiration in macrophages through the activation of eukaryotic translation initiation factor 5A-2[28]. In renal ischemia–reperfusion injury model mice, Spd reduced poly (ADP-ribose) polymerase 1 activation and DNA nitrative stress[29]. Although Spd was administered in both of these studies[28,29], Spd may be released from the injured tubules in the context of acute kidney injury. The biological defense response of increased spermidine levels in the impaired kidney may be more effective if it is supplemented by the administration of spermidine. The administration of Spd to 5/6 nephrectomized rats activated SIRT1 in vascular smooth muscle cells and reduced vascular calcification[30]. The protective effect of

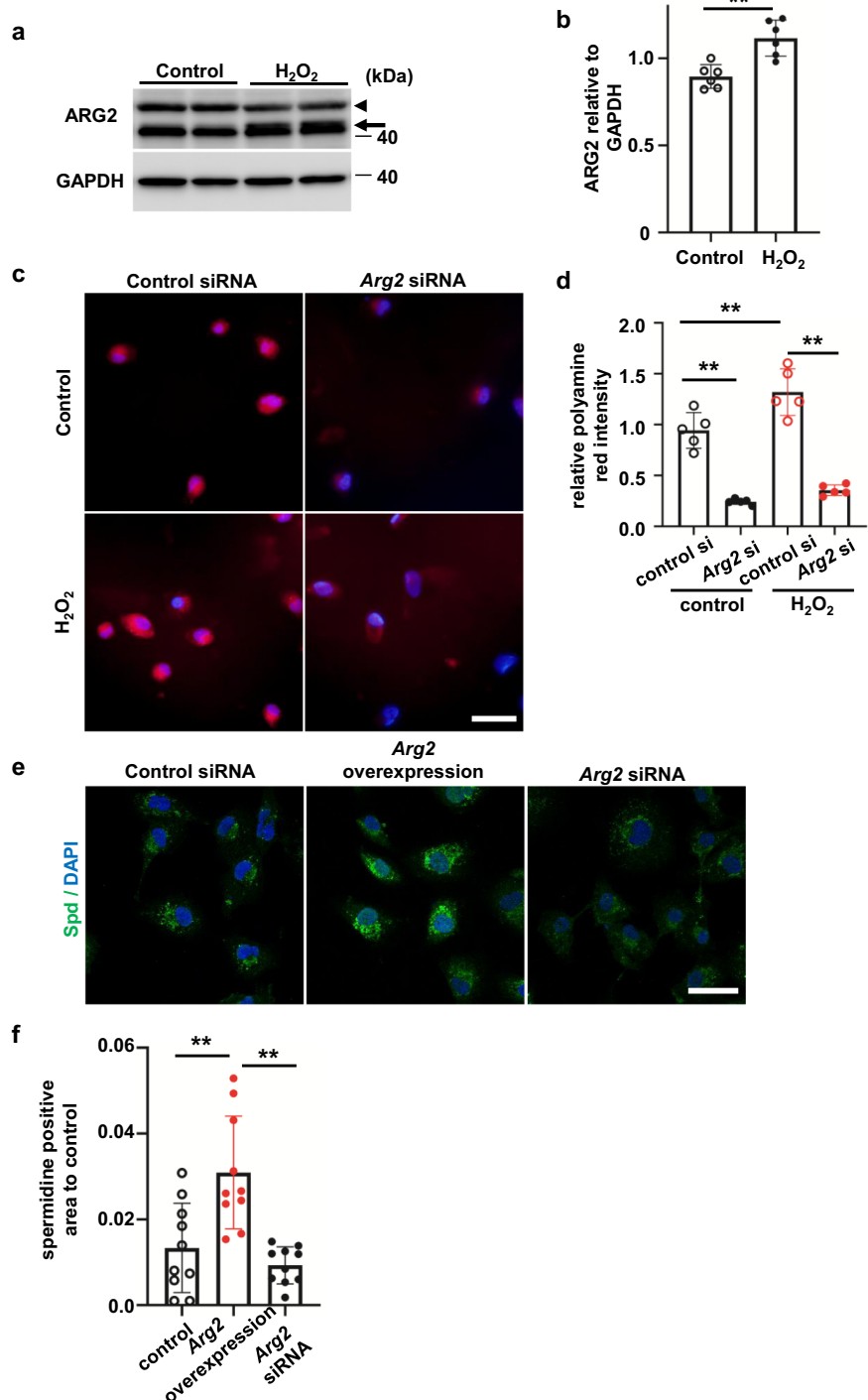

**Fig. 3 Oxidative stress upregulates ARG2 protein levels in renal tubular epithelial cells. a** Western blot analysis of ARG2 protein levels in HK-2 cells stimulated with $H_2O_2$. The arrowhead shows a nonspecific signal. **b** Relative levels of ARG2 protein normalized to GAPDH are shown ($n = 6$ in each group). Data are indicated as means ± SD. **c** Polyamine red staining in control or *Arg2* siRNA-transfected HK-2 cells stimulated with $H_2O_2$. Red, polyamine; blue, DAPI. Scale bars, 20 μm. **d** Quantification of polyamine red fluorescence intensity/cell shown in Fig. 3c ($n = 5$ in each group). Data are indicated as means ± SD. **e** Immunofluorescence images of Spd in HK-2 cells with *Arg2* overexpression or knockdown. Green, anti-Spd antibody; blue, DAPI. Scale bars, 20 μm. **f** Quantification of the Spd-positive area corrected for nuclei ($n = 10$ in each group). Data are indicated as means ± SD. *$P < 0.05$, **$P < 0.01$. Spd spermidine, ARG2 arginase 2, DAPI 4′,6-diamidino-2-phenylindole, GAPDH glyceraldehyde-3-phosphate dehydrogenase.

Spd on the kidneys has been observed in various cells of the kidney and may be more diverse than only transcriptional activation of Nrf2 via the activation of autophagy. We also found that acrolein protein levels were not reduced in *Arg2* KO mice (Supplementary Fig. S7d, e). These findings suggest that acrolein is produced from other pathways, including heat-induced dehydration of glycerol,

retro-aldol cleavage of dehydrated carbohydrates, lipid peroxidation of polyunsaturated fatty acids, and degradation of methionine and threonine[31], rather than from polyamines, in *Arg2* KO mice.

Nrf2 is a redox-sensitive transcription factor that regulates antioxidant proteins, cell cycle-related regulators, and detoxification enzymes[32]. Nrf2 has a renal protective role in various CKD

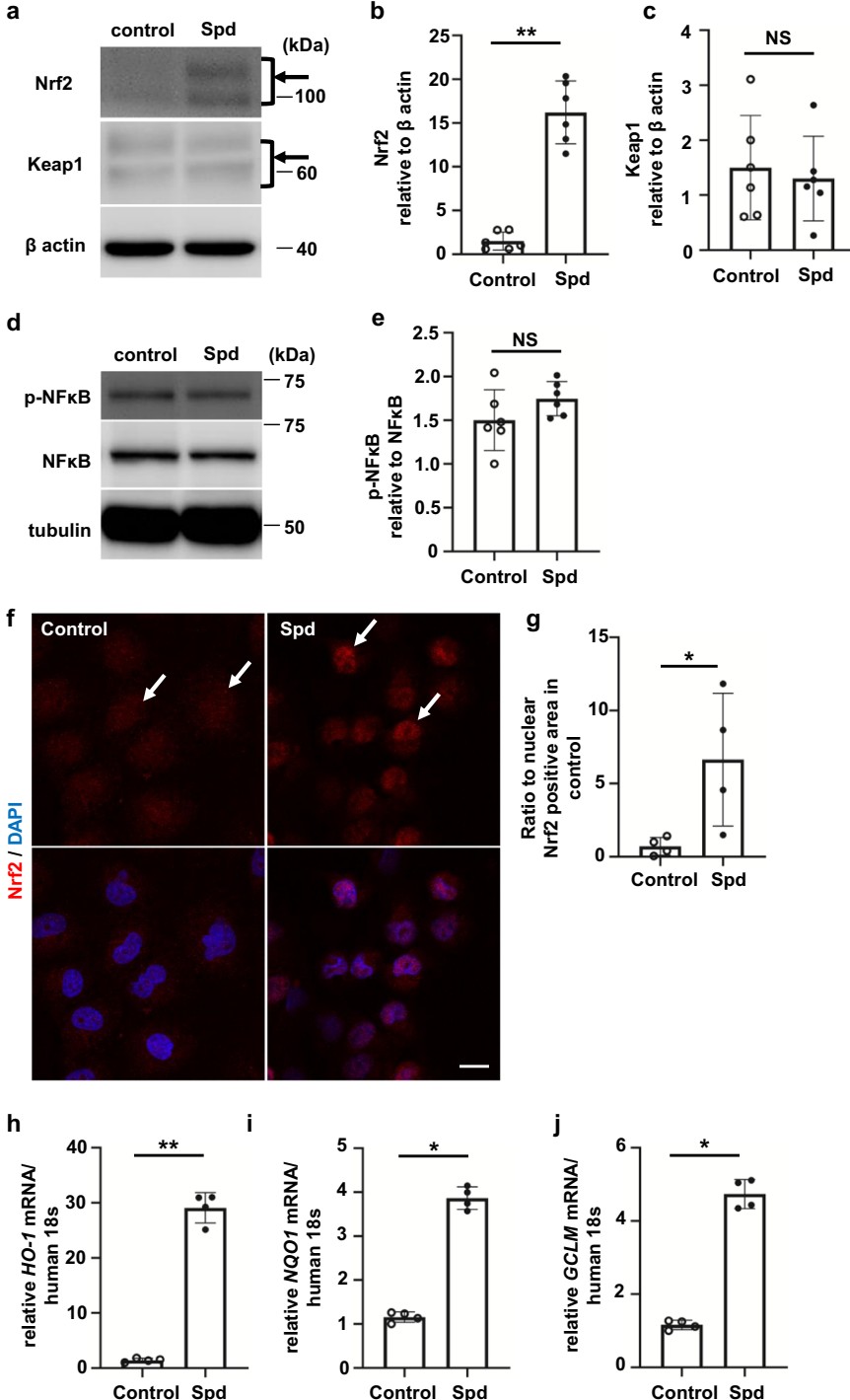

**Fig. 4 Spd activates the transcription factor Nrf2 in renal tubular epithelial cells. a** Western blot analysis of Nrf2 and Keap1 protein levels in HK-2 cells incubated with Spd. The arrow shows two Nrf2 bands. **b** Relative levels of Nrf2 protein normalized to β-actin are shown ($n = 6$ in each group). Data are indicated as means ± SD. **c** Relative levels of Keap1 protein normalized to β-actin are shown ($n = 6$ in each group). Data are indicated as means ± SD. **d** Western blot analysis of phospho- and total NF-κB in HK-2 cells incubated with Spd. **e** Relative levels of phospho-NF-κB protein normalized to total NF-κB are shown ($n = 6$ in each group). Data are indicated as means ± SD. **f** Immunofluorescence images of Nrf2 in HK-2 cells incubated with Spd. Red, anti-Nrf2 antibody; blue, DAPI. Scale bars, 20 μm. **g** Quantification of the nuclear Nrf2-positive area corrected for nuclei. Data are indicated as means ± SD. **h** HO-1, **i** NQO1, and **j** GCLM mRNA expression determined by real-time PCR in HK-2 cells incubated with Spd ($n = 4$ in each group). Data are indicated as means ± SD. *$P < 0.05$, **$P < 0.01$. Spd spermidine, Nrf2 nuclear factor erythroid 2-related factor 2, Keap1 kelch-like ECH-associated protein 1, NF-κB nuclear factor-κB, DAPI 4′,6-diamidino-2-phenylindole.

models[33–37]. Nrf2 deficiency promotes the progression from acute tubular damage to chronic kidney fibrosis following UUO[38]. The Nrf2 pathway suppresses TGFβ1-SMAD-mediated fibrosis signaling in HK-2 cells[39]. Intriguingly, previous studies have shown that

L-arginine induces an antioxidant response in the liver[40] or ameliorates cardiac oxidative stress via the Nrf2 pathway[41]. Although arginine supplementation activates Nrf2, whether arginine itself or arginine metabolites activate Nrf2 remains unclear. A higher

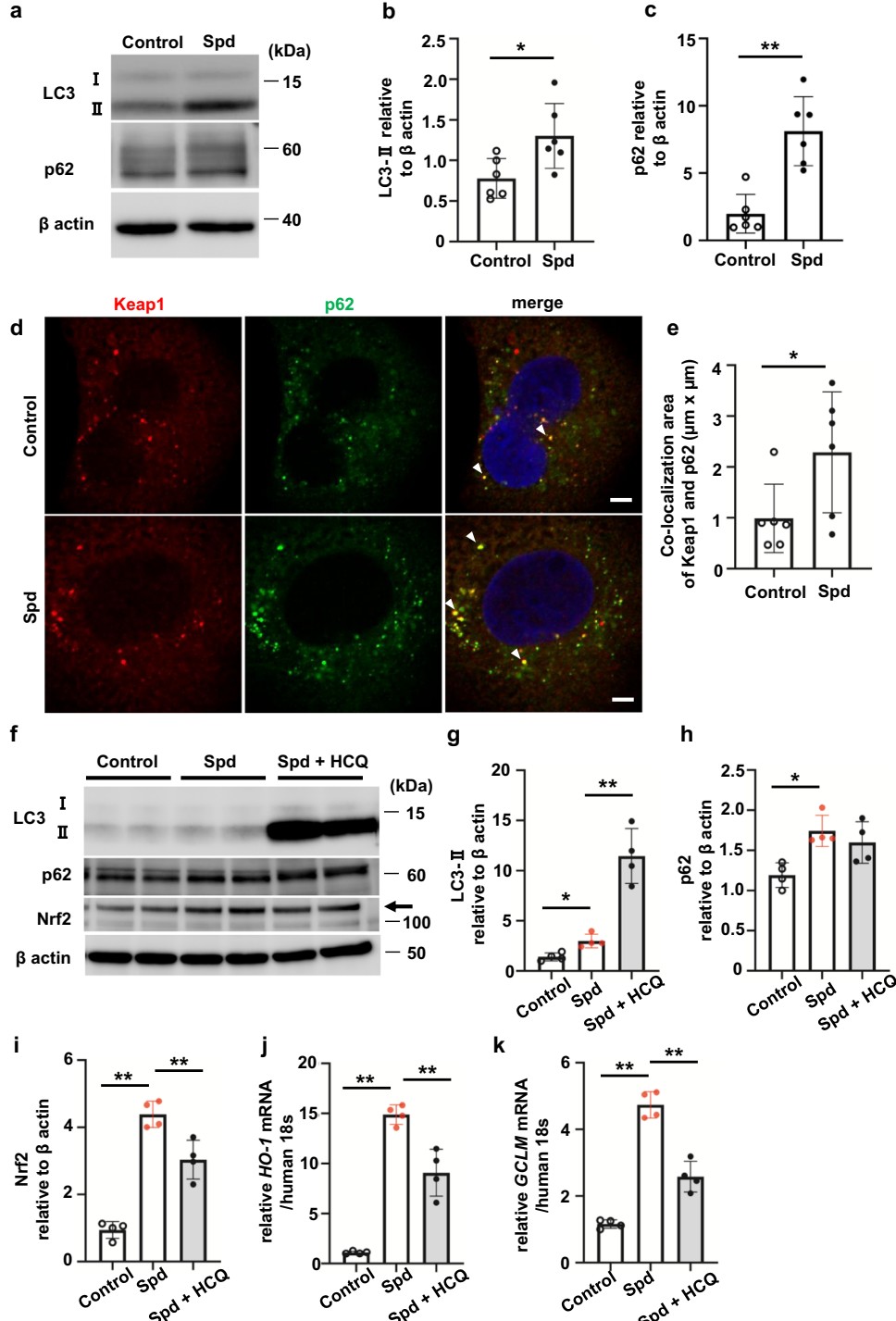

**Fig. 5 Spd activates autophagy, resulting in acceleration of Keap1 degradation in renal tubular epithelial cells. a** Western blot analysis of LC3-I and II and p62 in HK-2 cells incubated with Spd. **b** Relative levels of LC3-II protein normalized to β-actin are shown ($n = 6$ in each group). Data are indicated as means ± SD. **c** Relative levels of p62 protein normalized to β-actin are shown ($n = 6$ in each group). Data are indicated as means ± SD. **d** Confocal immunofluorescence microscopic images of Keap1 and p62 in HK-2 cells incubated with Spd. Red, anti-Keap1 antibody; green, anti-p62 antibody; blue, DAPI. White arrowheads indicate co-localized areas. Scale bars, 5 μm. **e** Quantification of co-localization areas of Keap1 and p62 in each cell. Data are indicated as means ± SD. **f** Western blot analysis of LC3-I and II, p62, and Nrf2 in HK-2 cells incubated with Spd in the presence of HCQ. **g** LC3-II, **h** p62, and **i** Nrf2 relative protein levels normalized to β-actin are shown ($n = 4$ in each group). Data are indicated as means ± SD. **j** *HO-1* and **k** *GCLM* mRNA expression determined by real-time PCR ($n = 4$ in each group). Data are indicated as means ± SD. *$P < 0.05$, **$P < 0.01$. Spd spermidine, LC3 microtubule-associated protein 1A/1B-light chain 3, Keap1 kelch-like ECH-associated protein 1, HCQ hydroxychloroquine.

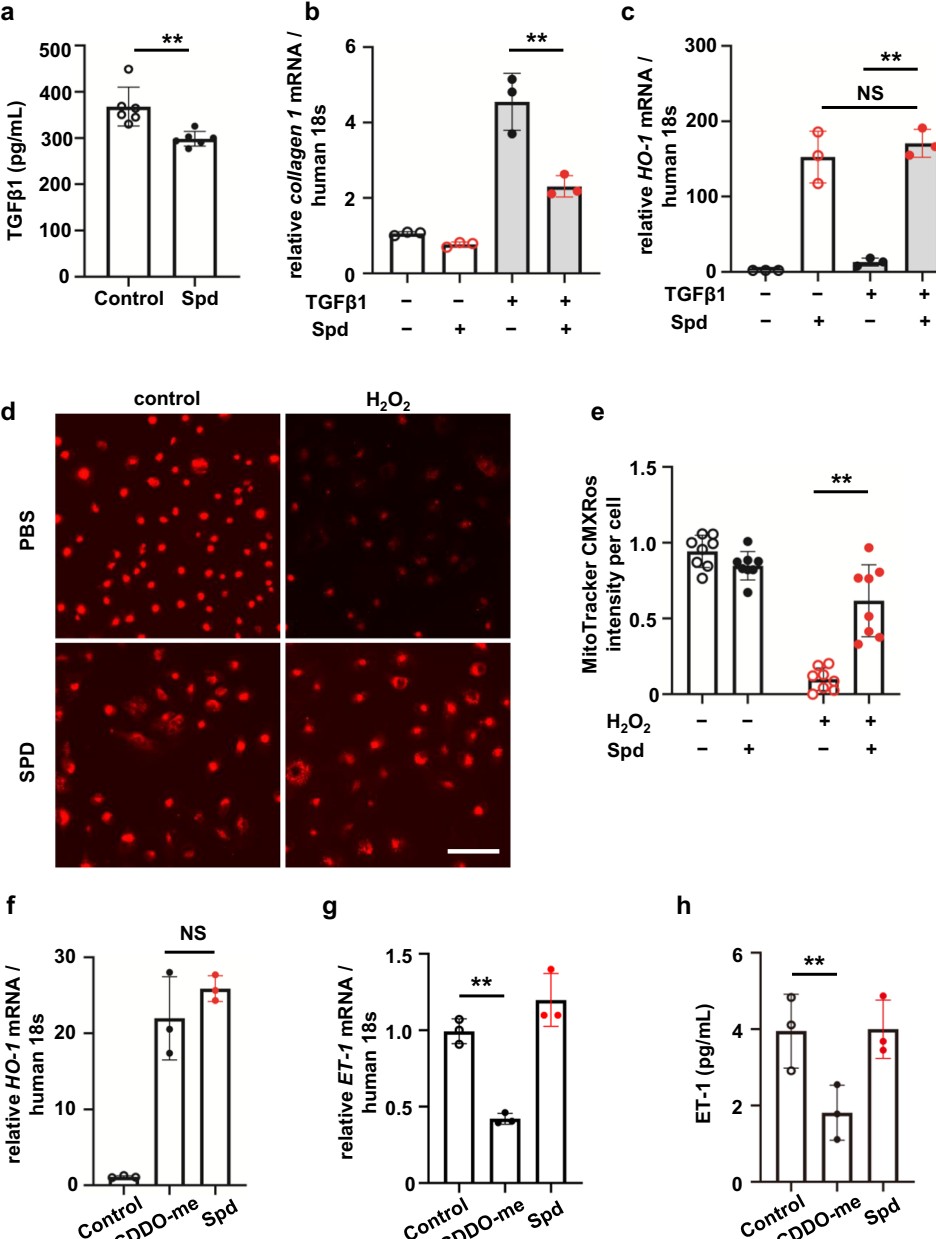

**Fig. 6 Spd suppresses fibrotic signaling, but does not inhibit the endothelin pathway like bardoxolone methyl. a** Secretion of TGFβ1 from HK-2 cells treated with Spd measured by ELISA ($n = 6$ in each group). Data are indicated as means ± SD. **b** Collagen 1 and **c** HO-1 mRNA expression determined by real-time PCR in HK-2 cells incubated with TGFβ1 in the presence of Spd ($n = 3$ in each group). Data are indicated as means ± SD. **d** MitoTracker Red CMXRos staining in HK-2 cells when exposed to $H_2O_2$ and Spd. Scale bar, 100 μm. **e** Quantification of Mitotracker Red CMXRos intensity measured by a multimode plate reader ($n = 8$ in each group). Data are indicated as means ± SD. **f** HO-1 and **g** ET-1 mRNA expression determined by real-time PCR in HK-2 cells incubated with Spd and CDDO-me ($n = 3$ in each group). Data are indicated as means ± SD. **h** Secretion of ET-1 from HK-2 cells treated with Spd and CDDO-me measured by ELISA ($n = 3$ in each group). Data are indicated as means ± SD. **$P < 0.01$. NS not significant, Spd spermidine, TGFβ1 transforming growth factor β1, CDDO-me bardoxolone methyl.

incidence of cardiovascular events with CDDO-me than with placebo prompted termination of a trial on advanced CKD[42]. Suppression of the endothelin pathway, which affects sodium and water homeostasis in the renal tubules, by CDDO-me is a mechanism contributing to adverse cardiovascular events[15]. The use of Spd in the treatment of CKD progression may avoid the issue of heart failure by activating Nrf2.

Spd is a natural polyamine involved in inducing autophagy[14]. The induction of autophagy in RTECs protects against apoptosis and promotes TGFβ degradation, thereby reducing development of renal interstitial fibrosis[43]. Spd also protects against liver fibrosis by enhancing Nrf2 signaling through activating autophagy[44]. In our study, co-localization of Keap1 and p62 was significantly increased by Spd in the cytoplasm of HK-2 cells (Fig. 5d, e). The binding of Keap1 to p62 was thought to increase free Nrf2. In a previous study, phosphorylation of p62 markedly increased p62's binding affinity for Keap1, and consequently, p62 phosphorylation induced expression of cytoprotective Nrf2 targets[45]. Another study showed that Nrf2 positively regulated p62 gene expression, which suggested a positive feedback loop[46]. These reports explain the phenomenon

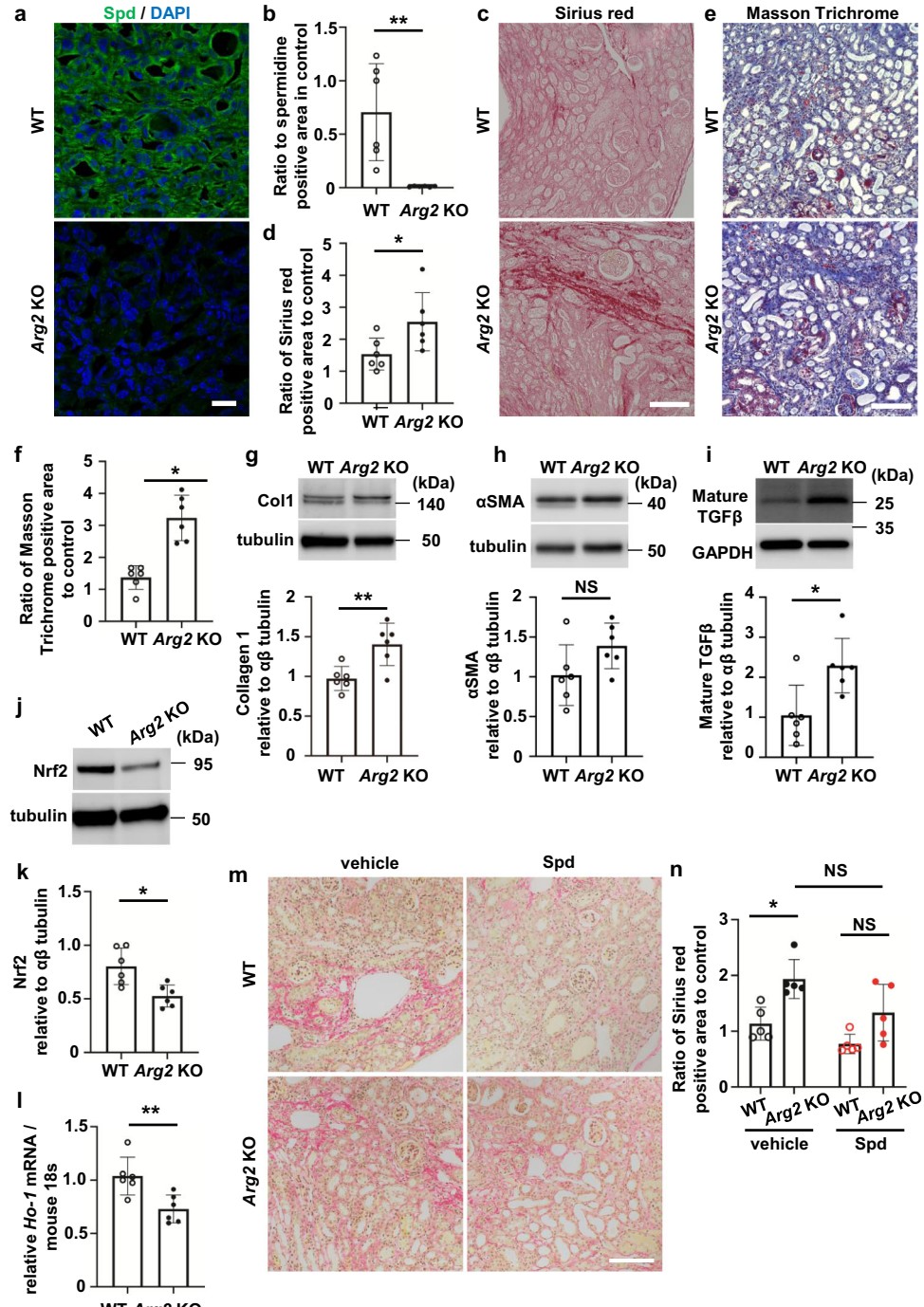

that p62 is increased by Spd, but its amount is not changed by autophagy inhibition in our experiments.

A limitation of this study is that Spd affects diverse pathways and cells, and effects on pathways other than the autophagy and Nrf2 pathways in tubular cells have not been investigated. Spd treatment of *Arg2* KO mice did not suppress fibrosis to WT levels. Therefore, we consider that the phenotype of *Arg2* KO mice may not be solely due to the amount of Spd.

In summary, arginine metabolism is activated and Spd is increased by fibrotic stimuli, resulting in acceleration of the Nrf2 pathway, which protects against interstitial inflammation and fibrosis in RTECs. Moreover, Spd activates the Nrf2 pathway in RTECs partly through the induction of autophagy. Increasing Spd levels may be effective in preventing chronic renal interstitial fibrosis.

## Methods

**Animal experiments**. To perform metabolomic analysis, 8-week-old C57BL/6Jcl mice were obtained from CLEA Japan Inc. (Tokyo, Japan). *Arg2* KO mice (*Arg2*<sup>tm1Weo</sup>/J) on a C57BL/6J background were purchased from Jackson Laboratory (Bar Harbor, ME, USA). Heterozygous mating was performed to generate WT and *Arg2* KO littermates. They were maintained in an air-conditioned, specific pathogen-free room at 21 °C and 65% humidity, with a 12-h:12-h light and dark cycle (lights on at 8:00 A.M., off at 8:00 P.M.) with free access to chow and water. All experimental protocols were approved by the Ethics Committee on Animal Experimentation, Kyushu University Graduate School of Medical Sciences (approval numbers: A19-294-0 and A21-093-0). Experiments were reported according to the ARRIVE guidelines.

**Experimental procedures for UUO in mice**. Mice were anesthetized intraperitoneally with medetomidine hydrochloride (0.3 mg/kg body weight; Orion Corporation, Espoo, Finland), 4 mg/kg midazolam (4 mg/kg body weight; Sandoz,

**Fig. 7 UUO-induced kidney fibrosis is aggravated in the *Arg2* knockout mouse kidney. a** Confocal immunofluorescence images of Spd in the UUO kidney of WT and *Arg2* KO mice. Green, anti-Spd antibody; blue, DAPI. Scale bars, 20 µm. **b** Quantification of the Spd-positive area ($n = 6$ in each group). Data are indicated as means ± SD. **c** Representative images of Sirius red staining in the UUO kidney of WT and *Arg2* KO mice. Scale bars, 100 µm. **d** Quantification of the Sirius red staining-positive area ($n = 6$ in each group). Data are indicated as means ± SD. **e** Representative images of Masson trichrome staining in the UUO kidney of WT and *Arg2* KO mice. Scale bars, 100 µm. **f** Quantification of the Masson trichrome staining-positive area ($n = 6$ in each group). Data are indicated as means ± SD. **g** Western blot analysis of collagen type 1 protein in the UUO kidney of WT and *Arg2* KO mice. Quantification of relative levels of collagen type 1 protein normalized to αβ tubulin are shown on the right ($n = 6$ in each group). Data are indicated as means ± SD. **h** Western blot analysis of αSMA protein in the UUO kidney of WT and *Arg2* KO mice. Quantification of relative levels of αSMA protein normalized to αβ tubulin are shown on the right ($n = 6$ in each group). Data are indicated as means ± SD. **i** Western blot analysis of mature TGFβ protein in the UUO kidney of WT and *Arg2* KO mice. Quantification of relative levels of mature TGFβ protein normalized to αβ tubulin are shown on the right ($n = 6$ in each group). Data are indicated as means ± SD. **j** Western blot analysis of Nrf2 protein in the UUO kidney of WT and *Arg2* KO mice. **k** Quantification of relative levels of Nrf2 protein normalized to αβ tubulin ($n = 6$ in each group). Data are indicated as means ± SD. **l** *HO-1* mRNA expression in the UUO kidney of WT and *Arg2* KO mice determined by real-time PCR ($n = 6$ in each group). Data are indicated as means ± SD. **m** Representative images of Sirius red staining in the UUO kidney of WT and *Arg2* KO mice treated with Spd. Scale, 100 µm. **n** Quantification of the Sirius red staining-positive area ($n = 5$ in each group). Data are indicated as means ± SD. *$P < 0.05$, **$P < 0.01$. NS not significant, UUO unilateral ureteral obstruction, Col1 collagen 1, Spd spermidine, DAPI 4′,6-diamidino-2-phenylindole, αSMA α smooth muscle actin, TGFβ transforming growth factor β1, Nrf2 nuclear factor erythroid 2-related factor 2.

Tokyo, Japan), and butorphanol tartrate (5 mg/kg body weight; Meiji Seika, Tokyo, Japan). Experimental unilateral ureteral ligation resulting in UUO was performed (on day 0) in 8-week-old-male mice by ligation of the left ureter of each mouse at the ureteropelvic junction. Experiments were conducted in UUO kidneys (WT, $n = 6$ and *Arg2* KO, $n = 6$). In Spd treatment experiments, five WT mice and five *Arg2* KO mice were used. The mice were intraperitoneally administered with Spd (10 mg/kg, S0266; Sigma-Aldrich, St. Louis, MO, USA), or phosphate-buffered saline (Nacalai Tesque, Kyoto, Japan) once a day from day 0 for 2 weeks as reported previously[47]. The body temperature of the mice was maintained at 37 °C during the whole procedure. Body weight was measured using an FX-3000 balance (A&D Company, Tokyo, Japan), and tail-cuff pressure was measured in a conscious state using a blood pressure monitor (MK-2000; Muromachi Kikai, Tokyo, Japan) before the operation and euthanasia.

**Sample collection**. Mice were euthanized on day 14 by intraperitoneal injection of 0.3 mg/kg medetomidine hydrochloride, 4 mg/kg midazolam, and 5 mg/kg butorphanol tartrate. Blood samples were collected from the inferior vena cava, aliquoted for later analysis, and stored at −80 °C. Immediately after blood collection, 50 mL of ice-cold phosphate-buffered saline (pH of 7.4) was slowly perfused to harvest both kidneys. Kidney samples were snap-frozen in liquid nitrogen and stored at −80 °C. Serum was separated by centrifugation at $2000 \times g$ for 10 min. Serum urea nitrogen and creatinine concentrations were determined using a dry-chemistry system (Fuji DRI-CHEM 7000VZ: Fujifilm Corp., Tokyo, Japan) and Fuji DRI-CHEM slides (Fujifilm Corp.).

**Metabolomic analysis of UUO kidney tissue**. Metabolomic analysis was performed at Human Metabolome Technologies, Inc. (Yamagata, Japan). Kidney lysate from C57BL/6Jcl mice with a sham operation or UUO for 14 days ($n = 4$ in each group) was used. Acetonitrile solution equivalent to 50% volume of tissue was added to the frozen kidney and homogenized at 4 °C. After centrifugation, the supernatant was transferred to an ultrafiltration tube (Ultrafree MC PLHCC, 5 kDa; Human Metabolome Technologies) and centrifuged. The filtrate was air dried and redissolved in 50 µL of MilliQ water for measurement. Capillary electrophoresis time-of-flight mass spectrometry was used to measure cationic metabolites, and capillary electrophoresis triple-quadrupole mass spectrometry was used to measure anionic metabolites. Capillary electrophoresis time-of-flight mass spectrometry measurement data were automatically extracted using MasterHands ver. 2.18.0.1 (developed by Keio University, Tokyo, Japan). With regard to capillary electrophoresis triple-quadrupole mass spectrometry measurement data, MassHunter (MassHunter Quantitative Analysis B.06.00; Agilent Technologies, Santa Clara, CA, USA) was used for automatic peak extraction.

**Histological examination and immunostaining**. Samples from both mouse kidneys were fixed in neutral-buffered 10% formalin (062-01661; Wako Pure Chemical Industries, Osaka, Japan) and embedded in paraffin. Four-micrometer sections were stained with Sirius red and Masson trichrome. The stained sections were obtained by an all-in-one fluorescence microscope (BZ-9000; Keyence, Osaka, Japan) and quantified in a blinded manner. Ten sections were randomly selected, and the percentage of kidney fibrosis was quantitated using Image J (National Institutes of Health, Bethesda, MD, USA) and Photoshop (Adobe Systems, San Jose, CA, USA). Mouse kidney tissues were prepared and stained as described previously[12]. The use of human kidney tissue and clinical data for analysis was approved by the ethics committee at Kyushu University (protocol # 28-385 for donor kidneys and #469-09 for IgA nephropathy). Human kidney specimens were obtained in Kyushu University Hospital from donor kidneys at the time of living donor kidney transplantation or from a renal biopsy specimen diagnosed as IgA nephropathy. The T scores indicating

tubular atrophy and interstitial fibrosis were semi-quantitatively classified in accordance with the Oxford Classification[48] of IgA nephropathy by the percentage of lesions in the cortical area as follows: T0 for 0%–25%, T1 for 26–50%, and T2 for >50%. We performed heat-based antigen retrieval using antigen activation solution (415211; Nichirei Biosciences Inc., Tokyo, Japan) and immunostained human kidney specimens embedded in paraffin with anti-Spd antibody.

To prepare for immunocytochemistry, HK-2 cells were seeded on Lab-Tek Chambered Glass Coverslips (154526; Thermo Fisher Scientific, Waltham, MA, USA) coated with fibronectin (F0895; Sigma-Aldrich). Cells were then fixed with 4% paraformaldehyde (09154-85; Nacalai Tesque) for 10 min at room temperature. Cells were permeabilized and blocked with 0.1% saponin (84510; Sigma-Aldrich) in 10% goat serum (426042; Nichirei biosciences) for 30 min. Primary antibodies were rabbit ARG2 polyclonal antibody (bs11397-R, 1:100; Bioss, Woburn, MA, USA) and rabbit Spd polyclonal antibody (ab7318, 1:100; Abcam, Cambridge, UK). The secondary antibody was Alexa 488-conjugated goat anti-rabbit IgG (A11008, 1:250; Thermo Fisher Scientific). After counterstaining with 4′,6-diamino-2-phenylindole, images were obtained by fluorescent microscopy (BX53; Olympus, Tokyo, Japan), the BZ-9000 digital microscope system (Keyence), or confocal laser scanning microscopy (LSM 700; Carl Zeiss, Oberkochen, Germany).

**Cell lines and culture**. Human renal proximal tubule (HK-2) cells were acquired from the American Type Culture Collection (Manassas, VA, USA). WT and *Atg5* KO mouse embryonic fibroblasts were obtained from Riken Cell Bank (Tsukuba, Japan). Cells were cultivated in Dulbecco's modified Eagle medium containing 10% fetal bovine serum (14A189; Sigma-Aldrich), 100 U/mL penicillin, and 100 mg/mL streptomycin (Thermo Fisher Scientific) in a humidified atmosphere with 5% $CO_2$ at 37 °C. The cells were cultured until 70% to 80% confluent in 6- or 12-well plates (Nippon Genetics, Tokyo, Japan) and then serum-deprived for 12 h before each experiment. The cells were stimulated with $H_2O_2$ (Nacalai Tesque) at 500 µM and Spd (Sigma-Aldrich) at 20 µM. ARG2 expression was suppressed by short interfering RNA (siRNA)-mediated knockdown. Control siRNA, *ARG2* siRNA, and *ATG5* siRNA were purchased from Horizon Discovery (Cambridge, UK). HK-2 cells were transfected with 10 nmol/L of the indicated siRNA using DharmaFECT 1 transfection reagent (Horizon Discovery). To perform ARG2 overexpression, the cells were transfected with pME18SFL3-human ARG2 (Toyobo, Osaka, Japan) using Lipofectamine 2000 (Thermo Fisher Scientific). To inhibit autophagy, the cells were incubated with 100 mM HCQ (H1306; Tokyo Chemical Industry, Tokyo, Japan). The cells were incubated with 10 ng/mL TGFß1 (Sigma-Aldrich) for fibrotic stimulus. The cells were treated with bardoxolone methyl (MedChemExpress, Monmouth Junction, NJ, USA) for comparison with Spd. To measure the viability of cells, Cell Count Reagent SF (#07553; Nacalai Tesque) was used. Cells were cultured in 96-well plates ($n = 6$) and 10 µL of reagent was added to each well. After incubation for 2 h, the absorbance at 450 nm was measured by a microplate reader (Molecular Devices, San Jose, CA, USA). Mitochondrial damage was assessed using MitoTracker Red CMXRos (Molecular Probes, Eugene, OR, USA). A volume of 100 µM MitoTracker Red CMXRos, which accumulates in mitochondria via the membrane potential, was added to live cells and incubated at 37 °C for 30 min. Stained cells were observed under a fluorescence microscope (BZ-9000; Keyence). All cell assays were performed in cells that had undergone five to eight passages.

**Western blotting**. Kidney tissues in lysis buffer (tissue protein extraction reagent with a protease inhibitor cocktail, 08714-04; Nacalai Tesque) containing a phosphatase inhibitor (06863-01; Nacalai Tesque) were homogenized twice for 3 min at 30 Hz using the TissueLyser (Qiagen, Hilden, Germany). The homogenate was centrifuged at $10,000 \times g$ for 10 min at 4 °C, and the collected supernatant was analyzed. The cells were harvested and lysed in cell lysis buffer (Mammalian Protein Extraction Reagent, #78501; Thermo Fisher Scientific) with protease

inhibitors or phosphatase inhibitors on ice for 15 min. The lysates were collected after centrifugation at $10,000 \times g$ for 10 min. Protein samples (10 µg) were separated by sodium dodecyl sulfate-polyacrylamide gel electrophoresis on 5% to 20% polyacrylamide gradient gels (2331830, PAGEL; Atto, Tokyo, Japan) and blotted onto a polyvinylidene difluoride membrane using the Trans-Blot Turbo System (BioRad, Hercules, CA, USA). Primary and secondary antibodies were diluted in antibody solution (signal enhancer HIKARI for western blotting and enzyme-linked immunosorbent assay [ELISA] Solutions A and B; Nacalai Tesque). After preincubation in blocking solution (Blocking One; Nacalai Tesque) for 30 min, the membranes were incubated overnight at 4 °C with the following primary antibodies: rabbit anti-ARG2 antibody (bs11397-R, 1:1000; Bioss); rabbit anti-SMOX antibody (15052-1-AP, 1:1000; Proteintech, Rosemont, IL, USA); rabbit anti-Nrf2 polyclonal antibody (ab137550, 1:1000; Abcam); rabbit anti-Keap1 polyclonal antibody (ab139729, 1:1000; Abcam); anti-NFκB polyclonal antibody (ab16502, 1:1000; Abcam); anti-phospho-NFκB polyclonal antibody (3033, 1:1000; Cell Signaling); mouse monoclonal anti-microtubule-associated protein 1A/1B-LC3 (CTB-LC3-2-IC, 1:1000; Cosmo Bio); anti-phospho-p62 antibody (95697 S, 1:1000; Cell Signaling Technology, Danvers, MA, USA); rabbit anti-collagen 1 antibody (ab34710, 1:5000; Abcam); mouse anti-αSMA antibody (ab7817, 1:300; Abcam); rabbit TGFβ antibody (CST3711, 1:1000; Cell Signaling Technology); mouse monoclonal acrolein antibody (MA5-27553, 1:1000; Thermo Fisher Scientific); rabbit arginase 1 antibody (CST9819, 1:1000; Cell Signaling Technology); mouse anti-eNOS/NOS type III (610297, 1:1000; BD Biosciences, San Jose, CA, USA); anti-CD31 antibody (ab28364, 1:500; Abcam); anti-Atg5 antibody (#12994; Cell Signaling Technology); rabbit anti-β-actin antibody (1:5000, ab8227; Abcam); rabbit anti-αβ-tubulin antibody (2148, 1:5000; Cell Signaling Technology); and mouse anti-glyceraldehyde-3-phosphate dehydrogenase (GAPDH) antibody (ab8245, 1:5000; Abcam). After being washed in tris-buffered saline with Tween 20 3 times, the membranes were incubated with the following horseradish peroxidase-conjugated secondary antibodies: anti-rabbit IgG antibody (NA934; GE Healthcare), anti-goat IgG antibody (sc-2033; Santa Cruz Biotechnology, Dallas, TX, USA), or anti-mouse IgG antibody (NA931; GE Healthcare, Chicago, IL, USA) for 1 h. The bands were detected using an enhanced chemiluminescent kit (11644-40; Chemi-Lumi One Ultra; Nacalai Tesque) and captured using a chemiluminescence imaging system (AE-9300 Ez-Capture MG; Atto). The density of each band was analyzed by ImageJ software (National Institutes of Health).

**Real-time polymerase chain reaction**. Total RNA was extracted from cells or tissues using the MAXWELL®16 LEV simplyRNA tissue kit (Promega, Madison, WI, USA) and the MAXWELL®16 instrument (Promega) according to the manufacturer's instructions. Complementary DNA was synthesized from 1 µg of total RNA with the PrimeScript RT Reagent kit (Takara Bio Inc., Shiga, Japan). Real-time polymerase chain reaction (PCR) was performed using SYBR Premix Ex Taq™ (Takara Bio Inc.) and the 7500 Real Time PCR System (Applied Biosystems, Foster City, CA, USA). The expression levels of each gene were calculated relative to the internal control. The primer sequences used for real-time PCR are shown in Supplementary Table S2.

**ELISA**. Cell supernatant was recovered and centrifuged at $3000 \times g$ for 10 min at 4 °C, and the collected supernatant was analyzed. To measure TGFβ1 and ET-1 concentrations in the supernatant, we used the human TGFβ1 Quantikine ELISA kit (DB100B; R&D Systems, Minneapolis, MN, USA) and the human ET-1 Quantikine ELISA kit (DET100; R&D Systems). To measure L-arginine concentrations in kidney tissue, we used the Mouse L-Arginine ELISA kit (MBS2600680; MyBioSource, San Diego, CA, USA).

**Statistics and reproducibility**. Parametric variables with a normal distribution are expressed as the mean ± standard deviation (SD). Parametric variables between two groups were compared using the Mann–Whitney $U$ test, and differences among groups were compared by one-way analysis of variance, followed by Tukey's post hoc test. Data with two independent variables were analyzed using two-way analysis of variance followed by Tukey's post hoc test. All statistical analyses were performed with EZR software (Saitama Medical Center, Jichi Medical University, Saitama, Japan), which is a graphical user interface for R (The R Foundation for Statistical Computing, https://www.r-project.org/), and a modified version of R commander designed to add statistical functions frequently used in biostatistics[49]. A two-tailed value of $P < 0.05$ was considered statistically significant. Trends in the mean values or frequencies of variables across subgroups were tested using the Jonckheere–Terpstra test or multiple linear regression analysis. We created graphs using GraphPad Prism version 6.0 for Windows (GraphPad Software, San Diego, CA, USA).

**Reporting summary**. Further information on research design is available in the Nature Portfolio Reporting Summary linked to this article.

## Data availability
The original data shown in the figures are available on Figshare, https://doi.org/10.6084/m9.figshare.23504841. Uncropped blots can be found in Supplementary Fig. S10. All other data are available from the corresponding authors on reasonable request.

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

## Acknowledgements
We thank the Research Support Center and Kyushu University Graduate School of Medical Sciences for technical support; M. Munakata and M. Tanaka at the Department of Medicine and Clinical Science, Graduate School of Medical Sciences, Kyushu University for assistance with histology; S. Kajiwara, a third-year student at Kyushu University School of Medicine, for ET-1 expression analysis; and Ellen Knapp, PhD, from Edanz (https://jp.edanz.com/ac) for editing a draft of this manuscript. This work was supported by a grant from the Japan Society for the Promotion of Science (Grant-in-Aid for Scientific Research: 20K08610).

## Author contributions
S. Aihara and K. Torisu conceptualized the study. S. Aihara, Y. Uchida, and N. Imazu acquired the data. S. Aihara interpreted the data, performed statistical analysis, and drafted the manuscript. K. Torisu provided intellectual content of the work. T. Nakano, K. Torisu, and T. Kitazono provided supervision, edited the manuscript, and obtained research funding.

## Competing interests
The authors declare no competing interests.
