## [Peer Review File · Communications Biology]

Reviewers' comments:

Reviewer #1 (Remarks to the Author):

The work by Aihara and colleagues is interesting and technically well done, at least in part. My main problem is that the claims made in the title and in the abstract are far too extensive. All conclusions concerning spermidines involvement in kidney fibrosis are based on in vitro data only.

The authors have two possibilities now:

1. repeat the experiments with spermidine supplementation (injection or feeding) in UOO mice (wild type and Arg2 knockout)

plus

- Inhibit spermidine production in UOO mice

or

2. Strongly reduce the claims (especially in the title)

In addition, the sentence: "In human proximal tubule cells, spermidine strongly induced nuclear factor erythroid 2 related factor 2 (Nrf2), and its induction was partly mediated by autophagy. " is an overstatement in the abstract, because the experiments have not been repeated in autophagy deficient cells.

Some Prior works should be cited and discussed:

Spermidine is protective against kidney ischemia and reperfusion injury through inhibiting DNA nitration and PARP1 activation

Aging Cell. 2021 Jun; 20(6): e13377.

Published online 2021 May 9. doi: 10.1111/accel.13377

Spermidine inhibits vascular calcification in chronic kidney disease through modulation of SIRT1 signaling pathway

Xiaoyu Liu, et al

Spermidine protects against acute kidney injury by modulating macrophage NLRP3 inflammasome activation and mitochondrial respiration in an eIF5A hypusination-related pathway
Molecular Medicine volume 28, Article number: 103 (2022) Cite this article

Reviewer #2 (Remarks to the Author):

This is a well prepared manuscript. The findings are novel and interesting. However, there are two minor points that need to be addressed.

(1) Statistical analysis: The data presented in Fig. 3d and 6b are derived from studies employing two independent variables. The data should be compared statistically by two-way ANOVA, with appropriate post hoc tests for differences among treatment groups. One-way ANOVA is appropriate for multiple levels of a single independent variable and fails to make full use of data derived from two or more independent variables. In general, statistical comparisons should be applied uniformly to the data obtained.

(2) The limitations of the current study should be discussed.

Reviewer #3 (Remarks to the Author):

In the present manuscript, Aihara et al. investigated the role of metabolite Spd in chronic renal interstitial fibrosis and the underlying mechanism. They show that arginine metabolism especially Spd level is enhanced in the UOO kidney in mice. Additionally, Spd treatment activates the transcription factor Nrf2 in HK-2 cells. Arg2 knockout reduces Spd levels, Nrf2 and HO-1 expression, and promotes UOO-induced kidney fibrosis. There are some concerns below.

1. Animal number and group information should be included in methods.

2. Please explain why the authors used 20 μM of Spd to treat cells.
3. It is better to list the primer sequences in a table.
4. Molecular weight is missing in WB.
5. The authors should provide the data showing the effect of Spd treatment on chronic renal interstitial fibrosis in UUO mice.
6. Did the authors examine the effect of Spd treatment on oxidative stress in HK-2 cells.
7. There are some typos and grammar mistakes in the manuscript. For example: "Spd expression was remarkably enhanced" should be "Spd level was remarkably enhanced".

We thank the reviewers for their useful comments and suggestions for our manuscript. We appreciate the reviewers' interest in our finding that Spd inhibits fibrosis of the kidneys. The reviewers have been important in shaping and strengthening our arguments during the revision process. We have performed additional experiments, as requested by the reviewers, and revised our manuscript in accordance with their suggestions as described below. Text in *italics* in this letter indicates the comments from the reviewers. The changes made in response to the reviewers' comments are shown in **red** in the revised manuscript. Text that has become unnecessary because of revision is shown in **blue** and has been deleted.

We therefore invite you to revise and resubmit your manuscript, taking into account the points raised. In particular we ask that you address all points raised by the 3 reviewers, specifically we ask that you address the statistical points raised by R2 and add more methodological information, as raised by R3. While the additional in vivo experiments suggested by R1 and R3 would be a plus, they are not mandatory, but, in their absence, please discuss the concomitant limitations of the study.

Response: The comments of the three reviewers are important and have been addressed as much as possible, including animal experiments.

Reviewer #1 (Remarks to the Author):

The work by Aihara and colleagues is interesting and technically well done, at least in part. My main problem is that the claims made in the title and in the abstract are far too extensive. All conclusions concerning spermidines involvement in kidney fibrosis are based on in vitro data only. The authors have two possibilities now:

Response: We thank the reviewer for the interest in our research. The title and abstract have been changed in accordance with the reviewer's remarks and the results of additional experiments.

1. repeat the experiments with spermidine supplementation (injection or feeding) in UOO mice (wild type and Arg2 knockout)

plus

- Inhibit spermidine production in UOO mice

Response: We agree with the reviewer that we should investigate whether administration of Spd can reduce fibrosis in the kidney. We injected Spd (10 mg/kg) intraperitoneally to UUU mice for 2 weeks as described in previous reports (PMID: 30241944, Gao M. et al, *Biochim. Biophys. Res. Comm.* 2018). Kidney fibrosis, which was significantly higher in *Arg2* KO mice than in WT mice, was suppressed by Spd, and this significant difference between WT and *Arg2* KO mice disappeared (Fig. 7m and n, Supplementary Fig. S9a). In the comparison between vehicle and Spd treatment, no significant difference in fibrosis was found in WT or *Arg2*KO mice, although the effect of Spd was greater in *Arg2* KO mice ($p = 0.38$ in WT mice and $p = 0.07$ in *Arg2* KO mice). We believe that the effects of Spd treatment were stronger in *Arg2* KO mice in which Spd levels were reduced.

An analysis of *Smox* KO mice would be informative in a mouse model that inhibits Spd production, but we have not performed this experiment. Therefore, we have described a previously reported AKI model of *Smox* KO mice and the expected phenotype in the CKD model in the Discussion section.

Changes have been made to the following parts of the manuscript:

Abstract: page 3, line 48–50

Introduction: page 5, line 86

Results: page 14, line 246 to page 15, line 254

Discussion: page 16, line 281, page 16, 284–page 17, line 289

Methods: page 21, lines 365–368

Fig. 7m, n

Supplementary Figure S9a

2. Strongly reduce the claims (especially in the title)

In addition, the sentence: “In human proximal tubule cells, spermidine strongly induced nuclear factor erythroid 2 related factor 2 (Nrf2), and its induction was partly mediated by autophagy. “is an overstatement in the abstract, because the experiments have not been repeated in autophagy deficient cells.

Response: The part of the title “...protects kidney from fibrosis” was replaced with “...inhibits kidney fibrosis”. HCQ alone could not explain the activation of Nrf2 from

Spd via autophagy. Therefore, *Atg5* knockdown tubular cells and *Atg5* KO mouse embryonic fibroblasts (MEFs) were examined. Nrf2 activation by Spd was also attenuated in *Atg5* knockdown cells (Supplementary Fig. S4b–d). In MEFs, unlike tubular cells, the activation of Nrf2 by Spd was not strong. However, Nrf2 activation by Spd was partially suppressed in *Atg5* KO MEFs. Autophagy is partly involved in Nrf2 activation, but it does not make a major contribution. Therefore, in the Abstract, we have removed our results indicating that Spd activates Nrf2 partly via autophagy.

Changes have been made to the following parts of the manuscript:

Title: page 1, lines 2

Abstract: page 3, lines 43, deleted

Results: page 11, lines 185–192

Discussion: page 19, line 334–338

Methods: page 25, lines 438–439, 447; page 28, lines 496

Supplementary Fig. S4b–g

Some Prior works should be cited and discussed:

Spermidine is protective against kidney ischemia and reperfusion injury through inhibiting DNA nitration and PARP1 activation

Aging Cell. 2021 Jun; 20(6): e13377.

Published online 2021 May 9. doi: 10.1111/acel.13377

Spermidine inhibits vascular calcification in chronic kidney disease through modulation of SIRT1 signaling pathway

Xiaoyu Liu, et al

Spermidine protects against acute kidney injury by modulating macrophage NLRP3 inflammasome activation and mitochondrial respiration in an eIF5A hypusination-related pathway

Molecular Medicine volume 28, Article number: 103 (2022) Cite this article

Response: We thank the reviewer for the suggestions. These three articles have been cited in the Discussion and discussed. These studies show that pathways other than Nrf2 and autophagy are also activated in Spd. Additionally, Spd has an effect on

macrophages and smooth muscle cells, as well as on tubular cells.

Changes have been made to the following parts of the manuscript:

Discussion: page 17, lines 291–302

References 28, 29, and 30, page 35, lines 618 – page 36, line 626

Reviewer #2 (Remarks to the Author):

This is a well prepared manuscript. The findings are novel and interesting. However, there are two minor points that need to be addressed.

(1) Statistical analysis: The data presented in Fig. 3d and 6b are derived from studies employing two independent variables. The data should be compared statistically by two-way ANOVA, with appropriate post hoc tests for differences among treatment groups. One-way ANOVA is appropriate for multiple levels of a single independent variable and fails to make full use of data derived from two or more independent variables. In general, statistical comparisons should be applied uniformly to the data obtained.

Response: Data with two independent variables, including those shown in Fig. 3d and 6b, were re-analyzed using two-way ANOVA. Additionally, post hoc tests of Tukey's multiple comparisons were conducted.

Changes have been made to the following parts of the manuscript:

Methods: page 30, 533–534

Figs. 3d, 6b, 6c, 6e, and 7n

Supplementary Figs. S4c, d, f, and g, and 9a

The above-mentioned figures show data that were analyzed using a two-way ANOVA.

(2) The limitations of the current study should be discussed.

Response: We have mentioned that the renoprotective effects of Spd are diverse and cannot be explained by our results alone. We have also added a sentence stating that the exacerbation of fibrosis in *Arg2* KO mice may not be due to a reduction in Spd alone (Discussion; page 19, lines 334–338).

Reviewer #3 (Remarks to the Author):

In the present manuscript, Aihara et al. investigated the role of metabolite Spd in chronic renal interstitial fibrosis and the underlying mechanism. They show that arginine metabolism especially Spd level is enhanced in the UUO kidney in mice. Additionally, Spd treatment activates the transcription factor Nrf2 in HK-2 cells. Arg2 knockout reduces Spd levels, Nrf2 and HO-1 expression, and promotes UUO-induced kidney fibrosis. There are some concerns below.

1. Animal number and group information should be included in methods.

Response: Information on the number of mice and experimental conditions has been added to the subsection “Experimental procedures for UUO in mice” in the Methods section (page 21, line 364 to 368). The conditions and numbers of mice used for metabolomic analysis are already described in the subsection “Metabolomic analysis of UUO kidney tissue” in the Methods section.

2. Please explain why the authors used 20 μ M of Spd to treat cells.

Response: The Spd concentration used in experiments is important. In previous reports, Spd was added to cultured cells, and concentrations varied from 0.1–200 μ M. We investigated cell viability at concentrations in this range in tubular cells (Supplementary Fig. S3a). On the basis of these results, 20 μ M Spd was selected as the concentration at which cell death does not occur. This information has also been added to the Results section.

Changes have been made to the following parts of the manuscript:

Results: page 9, lines 150–153

Methods: page 26, line 455–458

Supplementary Fig. S3a

3. It is better to list the primer sequences in a table.

Response: The primers have been listed in supplementary Table S2.

4. *Molecular weight is missing in WB.*

Response: Molecular weight markers were added as much as possible. There is also an uncropped membrane file, which will be submitted for the reviewers.

5. *The authors should provide the data showing the effect of Spd treatment on chronic renal interstitial fibrosis in UUU mice.*

Response: We agree with the reviewer that investigating whether administration of Spd can reduce fibrosis in the kidney is important. We injected Spd (10 mg/kg) intraperitoneally to UUU mice for 2 weeks as previously reported (PMID: 30241944, Gao M. et al, *Biochim. Biophys. Res. Comm.* 2018). Kidney fibrosis, which was significantly higher in *Arg2* KO mice than in WT mice, was suppressed by treatment with Spd, and the significant difference between WT and *Arg2* KO mice disappeared (Figure 7m and n, Supplementary Fig. S9). In the comparison between vehicle and Spd treatment, no significant difference in fibrosis was found in WT or *Arg2* KO mice, although the effect of Spd was greater in *Arg2* KO mice ($p = 0.38$ in WT mice and $p = 0.07$ in *Arg2* KO mice). We believe that the effects of Spd treatment were stronger in *Arg2* KO mice in which Spd levels were reduced.

Changes have been made to the following parts of the manuscript:

Abstract: page 3, line 48–50

Introduction: page 5, line 86

Results: page 14, line 246 to page 15, line 254

Discussion: page 16, line 281, 284– page 17, line 289

Methods: page 21, lines 365–368

Fig. 7m, n, Figure legend: page 46, line 831–833

Supplementary Figure S9a

6. *Did the authors examine the effect of Spd treatment on oxidative stress in HK-2 cells.*

Response: Whether Spd inhibits oxidative stress is an important question. The effect of Spd on mitochondrial damage was investigated by exposing HK-2 cells to H₂O₂. Oxidative stress, which is represented by a decrease in mitochondrial membrane potential caused by H₂O₂, was significantly suppressed by the addition of Spd (Fig. 6d and e).

Changes have been made to the following parts of the manuscript:

Results: page 12, lines 203–207

Methods: page 26, line 459–463

Fig. 6d, e

7. There are some typos and grammar mistakes in the manuscript. For example: “Spd expression was remarkably enhanced” should be “Spd level was remarkably enhanced”.

Response: We have changed “expression” to “levels” when referring to protein in the manuscript. We have had a native English speaker from a professional editing company correct any typos and grammatical errors.

REVIEWERS' COMMENTS:

Reviewer #1 (Remarks to the Author):

good revision

Reviewer #2 (Remarks to the Author):

The authors have properly addressed my questionings. I have no further concern on the revised version.

Reviewer #3 (Remarks to the Author):

I have no further comments.